# Effects of Both Japanese-Style Dietary Patterns and Nutrition on Falling Incidents among Community-Dwelling Elderly Individuals: A Cross-Sectional Study

**DOI:** 10.3390/nu14214663

**Published:** 2022-11-04

**Authors:** Ji-Woo Park, Satoko Kakuta, Rie Sakai, Tomoko Hamasaki, Toshihiro Ansai

**Affiliations:** 1Division of Community Oral Health Development, Kyushu Dental University, 2-6-1 Manazuru, Kokurakita-ku, Kitakyushu 803-8580, Japan; 2Division of Medical Nutrition Faculty of Healthcare, Tokyo Healthcare University, 3-11-3 Setagaya, Setagaya-ku, Tokyo 154-8568, Japan; 3Department of Nutrition, Faculty of Home Economics, Kyushu Women’s University, 1-1 Jiyugaoka, Yahatanishi-ku, Kitakyushu 807-8586, Japan

**Keywords:** older adults, community-dwelling, dietary behavior, mediterranean diet

## Abstract

Approximately 20% of the community-dwelling Japanese elderly (≥65 years) experience falling annually, with injury frequency rising with age. Increased nursing home admission/hospitalization risk influences healthy aging and QOL. Nutrition for musculoskeletal health is necessary, though the relationship of falling with nutritional status in the elderly is largely unknown. We investigated falling incidents and nutritional status, including a Japanese-style diet in a community-dwelling cohort. Using a cross-sectional design, 186 subjects (median age 83.0 years, males/females 67/119) were analyzed. Oral and systemic health conditions were assessed. A brief-type self-administered diet history questionnaire (BDHQ) was given for nutritional status. Analysis of covariance (adjusted for gender, age, BMI, articular disease/osteoporosis history, present tooth number, educational level) and the Japanese-Mediterranean diet (jMD) score adapted for Japan were used. The jMD score and falling incidents were significantly associated, with point increases related to a significantly decreased falling risk of 28% (OR: 0.72; 95%CI: 0.57–0.91). Of the 13 jMD food components, fish, eggs, and potatoes had a significant relationship with reduced falling, while significant associations of intake of animal protein, potassium, magnesium, zinc, and cholesterol (*p* < 0.05) were also observed. The results suggest that the jMD dietary pattern is an important factor for the prevention of falling incidents in elderly individuals.

## 1. Introduction

Individuals over 65 years old in Japan number 36.2 million and comprise 28.9% of the population. It has been reported that one in every three elderly people experience falling, with approximately 6% of those falls leading to fractures and 24% to severe injuries [1]. In addition, the frequency of falls is known to increase with age, peaking in the 80s. 

The annual incidence of falls at home among the elderly in Japan has been reported to range from 6.8% to 17.3% for men and 13.7% to 22.9% for women. Since bone health and muscle health are linked, sarcopenia, that is, age-related loss of muscle mass, strength, and function, should also be considered. Furthermore, osteosarcopenia can lead to a loss of independence and disability, as well as increased morbidity and mortality, and reduced quality of life [2].

Risk factors known to be associated with falling include advanced age, sarcopenia, osteoporosis, spinal arthritis and central nervous system disease, poor balance function, and depression [3]. Additionally, an association between oral status and falls has been reported [4], with a significantly increased risk of falling in individuals with 19 or fewer teeth and no dentures. Analyses of nutritional intake and bone health have also been summarized in several reviews [5]. Intake of vitamin D and protein, as well as calcium and magnesium, along with five servings of vegetables and fruits each day have been stated as important dietary recommendations for maintaining bone health. Furthermore, dietary patterns such as the Mediterranean diet (MED) are known to be associated with a lower risk of falling and fracture.

The MED was shown to be associated with a lower risk of falling in a Spanish report that studied older adults aged 60 years and older [6]. However, the association of the MED with falling incidents and fractures has yet to be established. According to a systematic review presented by Malmir et al. [7], there are few epidemiologic studies showing an association between the MED and bone mineral density or risk of fracture in older individuals in their 70s and 80s. Furthermore, though several related reports of subjects with similar ages have been presented, the results are likely biased due to the wide range of ages examined. On the other hand, a more recent epidemiological study of community-dwelling men aged 81 years found that adherence to the MED was associated with fewer falls, while consumption of higher levels of monounsaturated fatty acids and saturated fatty acids, and a higher ratio of monounsaturated fatty acids to saturated fatty acids, were associated with a reduced risk of falling [8]. In an epidemiological study of subjects with an average age of approximately 70 years, higher adherence to the MED was related to a lower risk of falling [6]. 

Previous studies presented findings that the consumption of animal protein led to greater muscle mass as compared to plant-sourced protein (e.g., soy and wheat) [9]. However, the details regarding the associations of the of plant- and animal-sourced diets with muscle and bone health were not clear [10]. The present study was conducted to investigate the effects of dietary patterns indicating nutrient intake on falling incidents in elderly individuals in Japan. We hypothesized that consumption of both animal protein and a Japanese-style dietary pattern by the elderly would lead to the prevention of falling incidents.

## 2. Materials and Methods 

### 2.1. Study Design and Participants

This was a cross-sectional study conducted from April 2015 to November 2020. It was conducted with residents of Buzen, Fukuoka Prefecture, Japan, a city with a population of about 25,000, who are mainly engaged in agriculture and fishing. All participants were enrolled in the National Health Insurance and the Late-Stage Senior Citizens Health Insurance plans.

### 2.2. Recruitment and Participants

The Buzen City government sent an invitation by mail to elderly individuals aged 75 years and older based on the population composition rate, as the Late-Stage Senior Citizens Health Insurance Plan in Japan provides coverage for citizens of that age. At the time of the initial invitation in April 2015, the percentage of males and females aged 75 or older was reported to be 35.3% and 64.7%, respectively. A total of 218 age-qualified individuals (34% male, 66% female) gave consent to the invitation and were surveyed by a home visitation. Of those, 186 (36% male, 64% female) were included in the present analysis, after exclusion of individuals with incomplete data, a daily energy intake <300 kcal, or who were bedridden (Figure 1). The present survey was approved by the Ethics Committee of our university (No. 20-31).

### 2.3. Sample Size Calculation

Sample size was estimated on the basis of a 30% prevalence of falling incidents by elderly individuals previously reported [1] and a relative 20% reduction in the non-faller group. Accordingly, a total of 156 individuals was estimated to achieve 80% power for a two-sided test at a statistical significance level of 5%.

### 2.4. Dietary Assessment

To assess intake of nutrients and food groups, a brief-type self-administered diet history questionnaire (BDHQ) was employed, which has been used in prior epidemiological studies, and confirmed for validity and accuracy by Kobayashi et al. [11]. The questionnaire asked respondents to indicate the frequency of consumption of specific foods among 58 food items, including beverages, by selecting from seven possible responses (twice a day or more, once a day, 4–6 times a week, 2–3 times a week, once a week, less than once a week, not eaten). This dietary survey method was designed to obtain information regarding an individual’s habitual intake of nutrients and foods during the past month’s meals [11]. Dietitians provided oral and written instructions on how to complete the BDHQ to the participants or their family members, and requested that they complete the form in advance. On the day of the home visit, a dietitian checked the BDHQ completed by the participant for errors or incompleteness, and also provided nutritional guidance, including food selection and cooking methods, as needed. For this study, the intake of each nutrient obtained from the BDHQ results was adjusted for energy using the density method (g/1000 kcal).

### 2.5. Japanese Modified Mediterranean Diet Score

According to the method reported by Kanauchi et al. [12], a Japanese-adapted MD (jMD) scoring system was used. Briefly, the jMD score was calculated based on the consumption of thirteen components (grains, vegetables, fruits, legumes, fish, dairy products, potatoes, poultry, eggs, red and processed meat, sweets, alcohol, ratio of monounsaturated to saturated fatty acids), with a value of 0 or 1 assigned to each of those thirteen components. One point was given for meeting each of the following consumption criteria; (1) three to seven servings of grains per day, (2) five or more servings of vegetables per day, (3) two or more servings of fruits per day, (4) 30 g or more of legumes per day, (5) 84 g or more of fish per day for men and 66 g or more for women, (6) 1.5–2.5 servings of dairy products per day, (7) potatoes consumed one to three times per week, (8) poultry two to three times per week, (9) eggs two to three times per week, (10) meat or meat products less than three times per week, (11) sweets less than two times per week, (12) 10–30 g of alcohol per day for men and 5–15 g for women, and (13) ratio of monounsaturated fatty acids to saturated fatty acids of 1.5 or greater. Otherwise, a score of zero was assigned. The range of possible total points was from 0 to 13, with a higher score indicating better adherence to the jMD.

### 2.6. Outcome Variable, Falling Incident

Incident falls were determined by the use of a short questionnaire. First the question “Have you had any falls over the previous 12 months?”, with a possible answer of “Yes” or “No”, was given. If “Yes” was answered, the subject was asked to report the number of falls. Those with more than two falls during that time were defined as “fallers”, as previous studies have found that single fallers are more similar to non-fallers than recurrent fallers in regard to a range of medical, physical, and psychological risk factors [13,14].

### 2.7. Body Composition

Body composition was assessed using bioelectrical impedance analysis with an InBody S10 device (InBody Japan, Tokyo, Japan). A tetra-polar eight-point tactile electrode system was also used, according to the manufacturer’s guidelines. Measurements were performed with the participant in a lying posture between 2:00 and 4:00 PM, with the immediate postprandial period avoided. Skeletal muscle mass (SMM) calculations were performed according to a previously reported method [15]. Briefly, the extremity skeletal muscle mass index (SMI) was determined based on the sum of the upper and lower extremities, then the SMI was calculated by dividing the SMM by participant height in square meters (kg/m^2^). 

### 2.8. Physical Function

A handgrip strength test was performed using a handheld dynamometer (T.K.K.5401, Takei Kiki Company, Niigata, Japan). Measurements were taken twice with each hand. After calculating the average value of each side, the larger value was used. Calf circumference (CC) on both sides was measured using a measuring tape with the largest circumference between the fibula head and outer edge of the lower leg noted.

### 2.9. Oral Examination

Assessments of tooth and/or denture conditions were performed by calibrated dentists, as described in our previous report [16].

### 2.10. Blood Examination

Blood tests were performed using a blood test kit (DEMECAL Leisure, Tokyo, Japan) [17]. Samples were collected from a fingertip and analyzed with a laboratory instrument (Demecal Healthcare Research Center, Yamanashi, Japan). The system provided results for 13 parameters; total protein, albumin, AST, ALT, γ-GTP, total cholesterol, HDL-C, triglycerides, BUN, creatinine, uric acid, glucose, and HbA1c.

### 2.11. Others

For medical history, information regarding hypertension, cardiovascular disease, cerebrovascular disease, cancer, dementia, articular disease, osteoporosis, and depression was obtained by letter from the primary care physician and the National Health Insurance database. Sarcopenia was assessed based on handgrip strength, CC, and SMI [15]. The Charlson Comorbidity Index, based on number of diseases noted in medical history records, was also calculated [18]. Information regarding alcohol consumption, smoking habit, and education level was obtained with a self-administered questionnaire.

### 2.12. Statistical Analyses

For comparisons between the faller and non-faller groups, a *t*-test was used when normality was observed, and the Mann–Whitney U-test was used when normality was not observed, with the Kolmogorov–Smirnov test used to test normality. When the distribution of nutrient intake did not follow a normal distribution, a log transformation was used to bring the analysis closer to a normal distribution. Mean differences between the groups were analyzed with analysis of covariance (ANCOVA), with covariates fed into the model selected based on previous studies. Risk factors related to falling were classified as internal or external [19]. Internal factors included age-related changes, such as muscle weakness, diseases such as arthritis, orthostatic hypotension, Parkinson’s disease, and hemiplegia, as well as medications taken, such as sleeping pills, antihypertensive drugs, and psychotropic drugs. External factors were considered to involve the living environment, such as house/apartment, clothing, and shoes. Other sociodemographic factors, such as educational history and assistive device use, in addition to gender and age, have also been reported to be associated [20]. Thus, the present model was adjusted for age, gender, body mass index (BMI), history of articular disease and/or osteoporosis, number of teeth, and educational level. All statistical analysis was executed using the SPSS software package, ver. 27 (IBM Corp., Armonk, NY, USA), with a *p* value < 0.05 considered to indicate significance.

## 3. Results

### 3.1. Basic Characteristics of Study Populations

The median age of the 186 subjects was 83.0 years, and there were 67 males and 119 females (36% and 64%, respectively). Table 1 shows the characteristics of the subjects according to falling status. Significant associations were observed in regard to history of articular disease (*p* = 0.02) and cancer (*p* = 0.04), while a significant tendency was observed between education level and falls (*p* = 0.05). No other significant association was found for the other variables.

### 3.2. JMD Food Intake and Falling

Table 2 shows the intake of each of the 13 food groups in the jMD with relationship to falling. Significant associations were found in regard to the intake of fish, eggs, and potatoes (*p* = 0.01, 0.03, and 0.04, respectively). All of the foods, except for grains, dairy products, and sweets, tended to be consumed more in the non-faller than faller group. 

### 3.3. Nutrient Intake and Falling

Table 3 shows nutrient intake according to falling incidents. Nutrients that still showed a reciprocal action after log transformation were excluded from the results. Significant associations were observed for the intake of protein (*p* = 0.01), animal protein (*p* < 0.01), potassium (*p* = 0.046), magnesium (*p* = 0.03), zinc (*p* = 0.02), and cholesterol (*p* < 0.01). Additionally, marginal associations were observed for iron, phosphorus, and vitamins D, E, and C.

### 3.4. Logistic Regression Analysis

Table 4 shows the results of logistic regression analysis of the association between the jMD and falling, with falls as the dependent variable. The results were adjusted for gender, age, and BMI (Model 1: [OR] 0.77, 95% [CI] 0.62–0.95, *p* = 0.016), and further for history of articular disease and osteoporosis (Model 2: [OR] 0.76, [CI] 0.61–0.94, *p* = 0.013). Model 3 shows the results after adjusting for the other covariates, plus number of teeth and education level ([OR] 0.72, [CI] 0.57–0.91, *p* = 0.005). The results indicated that a point increase in the jMD score was associated with a 28% decrease in falling risk. 

## 4. Discussion

The results of this cross-sectional study of community-dwelling elderly individuals showed that a reduction in risk of falling was significantly associated with the intake of potassium, magnesium, zinc, and cholesterol, as well as protein, including animal as opposed to plant protein. In addition to use of the jMD as a dietary pattern, which is consistent with findings described in a review by Rizzoli et al. [5], this is the first known study to examine the association of nutrients and dietary patterns with the risk of falling in very old individuals, and also the first such examination conducted with an Asian population. 

Dietary patterns have also been reported to be important for reducing fracture risk [5]. Other recent studies regarding the association of the MED as a dietary pattern with falls include systematic reviews presented by Ballesteros et al. [6] and Malmir et al. [7], both of which suggested that adherence to the MED lowered the risk of falling and fracture. Additionally, Ballesteros et al. [6] suggested that the accumulated or synergic impact of several foods, including fish and nuts, as well as vegetable intake, had a more beneficial effect as compared to a single food or nutrient. 

The involvement of bioactive components such as polyphenols, carotenoids, and phytosterols has been pointed out as a pathway by which the MED can influence sarcopenia and osteoporosis [21,22]. Thus, it is plausible that adherence to the jMD may be a factor related to reducing the risk of falling, as shown in the present study. 

On the other hand, there are some differences between the results of this and previous investigations. First, the present study found no association between vegetable intake and falling incidents, similar to the subgroup analysis findings presented by Lin et al. [23] of dark- and light-colored vegetables, though the reasons are unknown. Umezawa et al. [24] reported that the diet consumed in rural areas of Japan was characterized by higher intake of home-grown vegetables, as well as those given by neighboring farmers, as compared to such products purchased from a grocery store, likely due to cultural settings where vegetables are more readily available. This may explain why the present study found no significant difference between the faller and non-faller groups. Second, there was no significant association between calcium intake and falling shown in the present findings, while that between vitamin D and falling was only marginally significant. Many studies have reported that calcium intake is effective for reducing the risk of falls and fractures, though the optimal level of calcium remains to be firmly established. In particular, individuals over 75 years of age have decreased intestinal calcium absorption and the appropriate intake amount varies widely [5]. Lemming et al. [25] examined the associations of combinations of dietary calcium intake and the MED with the risk of hip fracture, and found that while calcium had some effect, calcium alone was not sufficient for the optimal prevention of a hip fracture. 

Nutrients found to be significantly associated with falling incidents included potassium, magnesium and zinc, which have positive effects on bone health, as described previously [26]. According to a recent review [5], magnesium has a role in parathyroid hormone secretion and potassium homeostasis. Another report noted that hip bone mineral density was positively correlated with magnesium intake [27]. Also, zinc is an essential nutrient used by various signaling pathways in nearly all cells in the human body, including bone cells. Details regarding the effects of these nutrients are discussed in the following. 

Based on the BDHQ, the food group from which each nutrient was derived was determined. For example, in the non-faller group, magnesium was found to be obtained from fish in 21.2% and vegetables in 21.8%. In other words, this population obtained magnesium from consuming both fish and vegetables. However, the results shown in Table 2 indicate that it was fish, eggs, and potatoes that were associated with falls, with most of the difference in magnesium consumption thought to be derived from fish. Results obtained the 2019 National Health and Nutrition Survey in Japan [28] indicated that 8.4% of magnesium was obtained from fish and 13.5% from vegetables, indicating that the present subjects consumed a much higher quantity of fish. This was consistent with our expectation based on the geographical location of the study area, which is facing the sea. Zinc is contained in oysters, and other shellfish, as well as meat and eggs. Similar to magnesium, a study using BDHQ found that about 49.2% of zinc consumed by a non-faller was derived from fish, meat, eggs, and dairy products, while 28.6% was from grains, and 8.9% from vegetables. These results may indicate potential reasons why fish and eggs were associated with falling incidents in our study. In present non-fallers, 32.0% of the potassium intake came from vegetables and 14.4% from fish. Although there was a significant difference in potassium intake between the non-faller and faller groups, vegetables, the main type of food consumed, were not found to be associated with falls for reasons mentioned earlier. Cholesterol and its metabolites have complex functions in osteogenesis, osteoclastogenesis, and bone homeostasis, depending on the form consumed [29]. Thus, the present findings indicate that cholesterol can sometimes be beneficial for bones. In addition, cholesterol is known to be included in large quantity in eggs, fish and shellfish, meat, and dairy products.

Though the differential effects of protein sources on musculoskeletal health remains to be clarified, potato protein supplementation was reported to increase muscle protein synthesis, both at rest and following resistance exercise in healthy young women [30]. On the other hand, there are few known epidemiological studies that have examined differences between animal and plant proteins, particularly in elderly individuals. The present results indicated a significant association of protein and animal protein intake with falling, while no association was seen with the intake of plant-sourced protein (Table 3). Among the 13 components of the jMD, there was a significant difference for potatoes. In general, the ingestion of animal protein leads to greater muscle mass as compared to plant-sourced protein, though there may not be a large difference between plant and animal proteins, especially among the elderly [10]. In the present results, grain ingestion was the most common (estimated at approximately 70%), while that of potatoes had a low percentage. This may explain why there was no significant difference regarding plant protein between the two groups. These results suggest that animal proteins have an important role in preventing against falling incidents, even in the elderly. Nevertheless, further studies are needed for confirmation.

The present subgroup analysis showed a gender difference regarding nutrients and fall risk, with a stronger trend in males (data not shown). Epidemiological studies that examined the association of the combined effect of dietary patterns and nutrition with the risk of falling have been limited, and most focused on a single gender. Although the reason for the difference between genders seen in the present results is unclear, the BDHQ mentioned that errors regarding food intake reported may vary between them [31], which might have influenced our results. Another possible reason is that the limited number of subjects did not provide sufficient statistical power. This study targeted community-dwelling individuals and is an ongoing project. It is important to provide community-based care, including dietary and nutritional guidance, on a regular basis through health education and information provided to local residents by collaborating with local government agencies to prevent falling incidents. 

Important aspects of the present results include the following. First, elderly individuals received focus, which has not been shown in previous studies. Second, the age range of the subjects was narrow and the cohort was homogenous. Furthermore, while associations of food components with the risks of falling and fracture have been examined, few epidemiological studies related to nutrients have been presented. In addition, there have been no studies regarding nutrients and the MED in Japanese individuals, except for the report by Kanauchi et al. [12]. 

This study had several limitations. Because of the cross-sectional nature, causality could not be analyzed. Furthermore, when using the BDHQ to examine nutrient intake in an epidemiological study, multiple surveys are desirable [11], while only a single BDHQ evaluation was used for the present analysis. In addition, some risk factors related to falling could not be investigated, as there was limited information regarding external factors, such as balance limiting activity, hearing impairment, or house structure, such as floors, lighting, and stepping obstacles like stairs. Finally, the enrolled subjects comprised approximately 3% of the elderly population aged 75 years or older in Buzen City; thus, generalizability is considered to be limited. The use of longitudinal cohorts with a larger sample size will be necessary to clarify the present results.

## 5. Conclusions

In the present study, the consumption of fish, eggs, and potatoes included in the diet were found to be associated with falling incidents in the elderly. Additionally, protein, especially animal protein, and the nutrients potassium, magnesium, zinc, and cholesterol were also associated with falls. Together, the results suggest that the jMD represents one of the healthiest dietary patterns and is an important factor for the prevention of falling incidents by elderly individuals.

## Figures and Tables

**Figure 1 nutrients-14-04663-f001:**
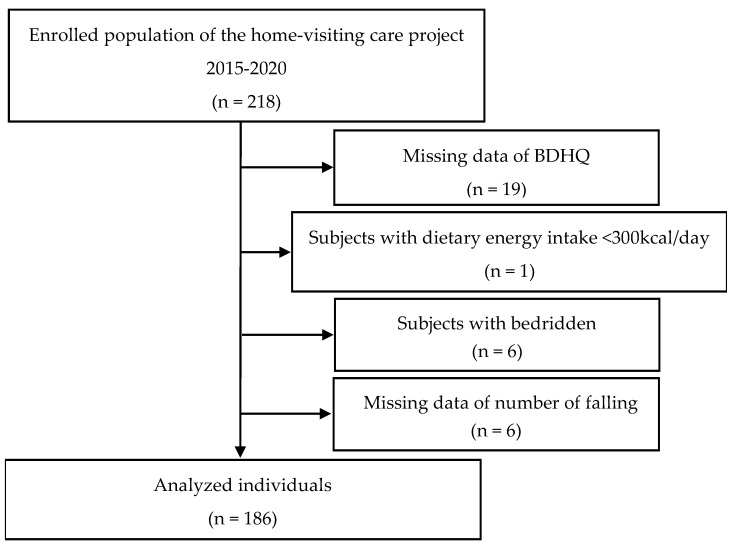
Flow diagram of participants selection.

**Table 1 nutrients-14-04663-t001:** Basic characteristics of study populations.

			Non-Faller *n* = 138		Faller *n* = 48		*p*

Systemic health status							
Age	Years		82.0 (76.0–88.0)		84.0 (80.0–88.0)		0.15 ^b^
Gender	Male (M)		48	(34.8)	19	(39.6)	0.55 ^c^
	Female (F)		90	(65.2)	29	(60.4)	
BMI	≥18.5 kg/m^2^		121	(87.7)	41	(85.4)	0.69 ^c^
	<18.5 kg/m^2^		17	(12.3)	7	(14.6)	
Energy intake	M	kcal/day	1979.1	(575.7)	1795.0	(416.0)	0.21 ^a^
	F	kcal/day	1527.0	(456.5)	1560.0	(505.7)	0.74 ^a^
SMI	M	<7.0 kg/m^2^	24	(51.1)	10	(55.6)	0.75 ^c^
	F	<5.7 kg/m^2^	49	(55.1)	15	(53.6)	0.93 ^c^
Hand grip strength	M	<28 kg	31	(66.0)	15	(78.9)	0.30 ^c^
	F	<18 kg	55	(61.1)	21	(72.4)	0.27 ^c^
Calf circumference	M	<34 cm	27	(56.3)	12	(63.2)	0.61 ^c^
	F	<33 cm	54	(60.0)	20	(69.0)	0.39 ^c^
Alb	g/dl		4.3	(4.0–4.4)	4.2	(3.9–4.4)	0.24 ^b^
Hypertension	*n* (%)		62	(44.9)	23	(47.9)	0.72 ^c^
Cardiovascular disease	*n* (%)		11	(8.0)	6	(12.5)	0.35 ^c^
Cerebrovascular disease	*n* (%)		21	(15.2)	10	(20.8)	0.37 ^c^
Dementia	*n* (%)		34	(24.6)	8	(16.7)	0.26 ^c^
Articular disease	*n* (%)		41	(29.7)	23	(47.9)	0.02 ^c^
Osteoporosis	*n* (%)		8	(5.8)	4	(8.3)	0.54 ^c^
Cancer	*n* (%)		22	(15.9)	2	(4.2)	0.04 ^c^
Depression	*n* (%)		3	(2.2)	1	(2.1)	0.97 ^c^
Sarcopenia	*n* (%)		51	(37.0)	21	(43.8)	0.41 ^c^
CCI	≥2 score		53	(38.4)	17	(35.4)	0.71 ^c^
	<2 score		85	(61.6)	31	(64.6)	
Oral health status							
Number of present teeth	≥20		63	(45.7)	22	(45.8)	0.98 ^c^
	<20		75	(54.3)	26	(54.2)	
Denture status (Upper)	Partial denture		42	(30.4)	16	(33.3)	0.91 ^c^
	Full denture		31	(22.5)	11	(22.9)	
	No denture		65	(47.1)	21	(43.8)	
Denture status (Lower)	Partial denture		43	(31.2)	18	(37.5)	0.67 ^c^
	Full denture		25	(18.1)	9	(18.8)	
	No denture		70	(50.7)	21	(43.8)	
Lifestyle factors							
Alcohol consumption	Yes		37	(26.8)	8	(16.7)	0.16 ^c^
	No		101	(73.2)	40	(83.3)	
Smoking status	Current smoker		4	(2.9)	1	(2.1)	0.86 ^c^
	Former smoker		36	(26.1)	11	(22.9)	
	Never smoker		98	(71.0)	36	(75.0)	
Social factors							
Education	Elementary school		17	(12.4)	2	(4.2)	0.05 ^c^
	Middle school		26	(18.8)	16	(33.3)	
	≥High school		95	(68.8)	30	(62.5)	

Abbreviations: BMI, body mass index; SMI, skeletal muscle mass index; Alb, albumin; CCI, Charlson comorbidity index. ^a^
*t*-test, ^b^ Mann–Whitney U test, ^c^ Chi-square test. Data indicate the number of subjects (%), with mean (SD) or median (IQR) shown in parentheses.

**Table 2 nutrients-14-04663-t002:** Association of jMD food group components with falling.

	Non-Faller		Faller		*p*
	*n* = 138		*n* = 48		
Food (g/1000 kcal) *	Mean (SD)		Mean (SD)		
Components					
Vegetables (g/day)	150.6	(78.7)	136.2	(66.8)	0.55
Fruits (g/day) ^a^	1.7	(0.34)	1.6	(0.40)	0.11
Legumes (g/day)	42.9	(24.8)	41.6	(25.4)	0.69
Fish (g/day)	59.3	(34.3)	48.2	(25.5)	0.01
Eggs (g/day)	27.6	(18.3)	20.8	(14.3)	0.03
Grains (g/day) ^b^	221.2	(66.6)	234.5	(71.3)	0.21
Poultry (g/day) ^a^	1.1	(0.3)	1.1	(0.3)	0.32
MUFA/SFA (ratio) ^c^	1.4	(0.2)	1.3	(0.3)	0.66
Potatoes (g/day) ^a^	1.5	(0.5)	1.3	(0.4)	0.04
Dairy products (g/day)	64.4	(54.9)	78.5	(59.6)	0.45
Red and processed meat (g/day)	23.2	(13.0)	20.5	(12.3)	0.16
Sweets (g/day)	25.2	(21.6)	28.3	(26.6)	0.64
Alcohol (g/day)	1.4	(4.8)	0.9	(3.9)	0.47

* All intake was adjusted for energy intake with a density method. ^a^ All variables were log-transformed before analyses. ^b^ Grains included consumption of cereals, noodle, bread and rice (whole and refined grains). ^c^ MUFA/SFA (ratio), mono unsaturated fatty acids (g/day)/saturated fatty acids (g/day). ANCOVA, adjusted for gender, age (<80/≥80 years), body mass index (continuous), history of articular disease (yes/no), history of osteoporosis (yes/no), number of present teeth (<20/≥20), and education level (elementary/middle/high school).

**Table 3 nutrients-14-04663-t003:** Association of nutrient intake with falling.

	Non-Faller		Faller		*p*
	*n* = 138		*n* = 48		
Food (g/1000 kcal) *	Mean (SD)		Mean (SD)		
Nutritional component					
Energy (kcal/day)	1684.3	(543.9)	1653.0	(481.8)	0.70
Protein (g/day)	42.7	(9.1)	39.3	(7.3)	0.01
Animal protein (g/day)	26.0	(9.8)	22.4	(7.8)	<0.01
Plant protein (g/day)	16.7	(2.7)	16.9	(2.0)	0.24
Fat (g/day)	28.7	(6.0)	27.9	(6.0)	0.61
Carbohydrate (g/day)	138.2	(19.4)	143.1	(18.5)	0.07
Sodium (mg/day)	2441.8	(621.6)	2397.1	(472.6)	0.40
Potassium (mg/day)	1584.6	(434.0)	1442.8	(388.3)	<0.05
Calcium (mg/day)	328.9	(106.2)	319.8	(107.9)	0.34
Magnesium (mg/day)	152.4	(34.6)	140.0	(30.9)	0.03
Iron (mg/day)	5.1	(1.3)	4.6	(1.2)	0.05
Phosphorus (mg/day)	647.5	(145.6)	595.4	(133.5)	0.07
Vitamin D (μg/day)	10.9	(5.5)	9.4	(5.3)	0.08
Vitamin E (mg/day)	4.3	(1.1)	4.0	(1.0)	0.07
Vitamin C (mg/day)	86.0	(36.6)	76.1	(34.3)	0.06
Folic acids (μg/day)	230.7	(78.0)	210.3	(74.7)	0.15
Zinc (mg/day)	4.9	(0.8)	4.5	(0.7)	0.02
Cholesterol (mg/day)	249.8	(97.5)	206.4	(84.0)	<0.01

* All intake was adjusted for energy Intake with a density method. ANCOVA, adjusted for gender, age (<80/≥80 years), body mass index (continuous), history of articular disease (yes/no), history of osteoporosis (yes/no), number of present teeth (<20/≥20), and education level (elementary/middle/high school).

**Table 4 nutrients-14-04663-t004:** Multivariable adjusted OR and 95% CI for the association of jMD and falling.

Variables	Crude Model OR (95% CI)	*p*	Model 1 OR (95% CI)	*p*	Model 2 OR (95% CI)	*p*	Model 3 OR (95% CI)	*p*
jMD (per one point increase) (continuous)	0.78 (0.63–0.96)	0.017	0.77 (0.62–0.95)	0.016	0.76 (0.61–0.94)	0.013	0.72 (0.57–0.91)	0.005
Female (male: Ref)			0.67 (0.33–1.36)	0.26	0.67 (0.33–1.37)	0.27	0.64 (0.31–1.36)	0.25
Age (≥80) (<80: Ref)			2.19 (1.01–4.74)	0.046	1.88 (0.85–4.16)	0.12	2.59 (1.12–6.02)	0.027
BMI (kg/m^2^) (continuous)			1.02 (0.93–1.12)	0.62	1.02 (0.93–1.12)	0.70	1.02 (0.93–1.13)	0.62
Articular disease					2.02 (0.98–4.16)	0.06	2.15 (1.00–4.59)	0.049
Osteoporosis					0.97 (0.26–3.70)	0.97	1.29 (0.32–5.22)	0.72
Number of present teeth (<20) (≥20: Ref)							0.52 (0.24–1.13)	0.10
Education level (elementary) (middle) (≥high school: Ref)							0.21(0.04–1.03) 1.81(0.81–4.03)	0.05
								0.15


Model 1: logistic model adjusted for gender, age and BMI. Model 2: logistic model with additional adjustments for articular disease, osteoporosis. Model 3: logistic model with additional adjustments for number of present teeth, education level. Ref: reference, OR: odds ratio, CI: confidence interval.

## Data Availability

The datasets analyzed during the current study are available from the corresponding author upon reasonable request.

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
