# Peer review of "Effects of Both Japanese-Style Dietary Patterns and Nutrition on Falling Incidents among Community-Dwelling Elderly Individuals: A Cross-Sectional Study"

_nutrients, 2022, doi:10.3390/nu14214663_

Round 1

Reviewer 1 Report

This is an interesting study that relates the intake of nutrients in a Japanese population, evaluated with the administration of an evaluation scale built following the model of the Mediterranean Diet. The result is interesting, as it shows an inverse relationship between good nutritional status and the risk of falling, conducted on an Asian population.

Author Response

Following are our replies to the comments from Reviewer I

This is an interesting study that relates the intake of nutrients in a Japanese population, evaluated with the administration of an evaluation scale built following the model of the Mediterranean Diet. The result is interesting, as it shows an inverse relationship between good nutritional status and the risk of falling, conducted on an Asian population.

  • We appreciate the helpful comments from the reviewer.

Reviewer 2 Report

-Abstract: to add numbers in the results section.

-Abstract: what is correct conclusion?

-Introduction: Are protein vegetable essential to fall and muscle? Please, clarify.

-Introduction: What is hypothesis?

-Methods: What are advices to do bioelectrical impedance analysis? Fasting or not?

In addition, muscle mass was calculated using formulae? or values were acquired from BIA? what is formulae?

-Results: Table 3 did not show association, but consumption only.

In addition, animal ingestion  is inversely associated with fall?

-Discussion and conclusion: Does not make sense since protein and alkaline-producing dietary, such as potassium, magnesium, zinc. What kind of protein? vegetable? animal? this is very controversy and must be re-writting and discussed. 

Author Response

Following are our replies to the comments of the Reviewer II:

  1. Abstract: to add numbers in the results section.
    • Accordingly, statistical values have been added to the Abstract in the revised manuscript (P.1, lines 26 and 29).
  2. Abstract: what is correct conclusion?
    • The description regarding the conclusion of the study presented in the Abstract has been rewritten for clarity (P. 1, lines 29 to 30).
  3. Introduction: Are protein vegetable essential to fall and muscle? Please, clarify.
    • Recently, there has been debate regarding differences between animal- and plant-based proteins. An explanation in that regard has been added (P.2, lines 71 to 79).
  4. Introduction: What is hypothesis?
    • To answer this question from the reviewer, a sentence regarding our hypothesis based on the results has been added (P.2, lines 81 to 83).
  5. Methods: What are advices to do bioelectrical impedance analysis? Fasting or not?
    • In the present study, measurements were not taken under a fasting condition, though the immediate postprandial period was avoided. An explanation has been added to the Materials and Methods section in the revised manuscript (P.4, lines 148 to 150).
  6. In addition, muscle mass was calculated using formulae? or values were acquired from BIA? what is formulae?
    • BIA with an InBody S10 was used for this study. The Asian Working Group for Sarcopenia (AWGS) has accumulated a large amount of evidence regarding sarcopenia using that device and studies conducted with Asians are increasing, as noted in a recent article (Chen L-K, et al., JAMDA, 2020). A related sentence has been added to the Materials and Methods section (P.4, lines 150 to 153).
  7. Results: Table 3 did not show association, but consumption only.
    • Table 3 shows results of analysis of covariance (ANCOVA), a statistical analysis method, as described in the Materials and Methods section (P.5, lines 186 to 188).
  8. In addition, animal ingestion is inversely associated with fall?
    • As the reviewer pointed out, our results show a significant association between animal protein consumption and risk of falling. Our interpretation of these results is presented in the Discussion section (P. 11, lines 326 to 331).
  9. Discussion and conclusion: Does not make sense since protein and alkaline-producing dietary, such as potassium, magnesium, zinc. What kind of protein? vegetable? animal? this is very controversy and must be re-writing and discussed. 
    • We appreciate these important comments and questions. Related portions of the Introduction and Discussion section have been revised (P.2, lines 71 to 79; P.11, lines 323 to 331). The aim of the study was to verify that animal protein consumption contributes to prevention of falling incidents, even in elderly individuals. In the Introduction section, an explanation of which protein source, animal or plant, has a more positive impact on muscle health is presented (P.2, lines 71 to 78).

Round 2

Reviewer 2 Report

Discussion and conclusion: was little answered by authors. Thus, my question continues:

Does not make sense since protein and alkaline-producing dietary, such as potassium, magnesium, zinc. What kind of protein? vegetable? animal? this is very controversy and must be re-writing, discussed and concluded.

Author Response

Following are our replies to the comments of Reviewer II:

  1. Discussion and conclusion: Does not make sense since protein and alkaline-producing dietary, such as potassium, magnesium, zinc. What kind of protein? vegetable? animal? this is very controversy and must be re-writing, discussed and concluded. 
    • We appreciate these important comments and questions. Accordingly, the entire manuscript has been revised, with focus on the Introduction, Discussion, and Conclusion sections. The following conclusions were made based on our findings. (1) The Japanese-adapted Mediterranean diet (jMD) is one of the healthiest and can play a role for prevention of falling incidents. (2) The present non-faller group consumed more animal protein, including fish and eggs, as compared to the faller group.
    • Details regarding revisions made to the manuscript are shown following:
    • Abstract section, lines 29 to 31. “The results suggest that the jMD dietary pattern is an important factor for prevention of falling incidents in elderly individuals.”

Introduction section, lines 71 to 74. Sentences regarding current debate related to differences between plant- and animal-based proteins have been added, as follows. “Previous study presented findings that consumption of animal protein leads to greater muscle mass as compared to plant-sourced protein (e.g., soy and wheat) [9]. However, the details regarding the associations of the of plant- and animal-sourced diets with muscle and bone health are not clear [10].

  • Discussion section. Discussion regarding our findings in association with several previous reports regarding dietary pattern including the MED has been added, including several sentences related to nutrients such as animal protein, potassium, magnesium, and zinc. In addition, text has been included to show our interpretation based on BDHQ outcomes (Page 10, lines 341 to 349, and Page 11, lines 350 to 370).